# Achilles Scraping and Plantaris Tendon Removal Improves Pain and Tendon Structure in Patients with Mid-Portion Achilles Tendinopathy—A 24 Month Follow-Up Case Series

**DOI:** 10.3390/jcm10122695

**Published:** 2021-06-18

**Authors:** Lorenzo Masci, Bradley Stephen Neal, William Wynter Bee, Christoph Spang, Håkan Alfredson

**Affiliations:** 1Institute of Sports Exercise and Health, University College Hospital London, London W1T 7HA, UK; lorenzo@sportdoctorlondon.com (L.M.); williamwb@doctors.org.uk (W.W.B.); hakan.alfredson@umu.se (H.A.); 2Sports & Exercise Medicine, Queen Mary University of London, London E1 4DG, UK; b.s.neal@qmul.ac.uk; 3School of Sport, Rehabilitation and Exercise Sciences, University of Essex, Wivenhoe Park, Colchester CO4 3SQ, UK; 4Department of Integrative Medical Biology, Anatomy Section, Umeå University, 901 87 Umeå, Sweden; 5Private Orthopaedic Spine Center, 97080 Würzburg, Germany; 6Department of Community Medicine and Rehabilitation, Sports Medicine, Umeå University, 901 87 Umeå, Sweden; 7Pure Sports Medicine, Canary Wharf, London E14 4QT, UK

**Keywords:** Achilles tendon, tendinopathy, plantaris, surgery, after treatment, follow-up study

## Abstract

Background: Studies have demonstrated that a sub-group of patients with medial Achilles pain exhibit Achilles tendinopathy with plantaris tendon involvement. This clinical condition is characterised by structural relationships and functional interference between the two tendons, resulting in compressive or shearing forces. Surgical plantaris tendon removal together with an Achilles scraping procedure has demonstrated positive short-term clinical results. The aim of this case series was to determine the long-term outcomes on pain and Achilles tendon structure. Methods: 18 consecutive patients (13 males; 5 females; mean age 39 years; mean symptom duration 28 months), of which three were elites, were included. Clinical examination, b-mode ultrasound (US) and Ultrasound Tissue Characterisation (UTC) confirmed medial Achilles tendon pain and tenderness, medial Achilles tendinopathy plus a plantaris tendon located close to the medial side of the Achilles tendon. Patients underwent US-guided local Achilles scraping and plantaris tendon removal followed by a structured rehabilitation program. Outcomes were VISA-A score for pain and function and UTC for Achilles structure. Results: 16 of 18 patients completed the 24 months follow-up. Mean VISA-A scores increased from 58.2 (±15.9) to 92.0 (±9.2) (mean difference = 33.8, 95% CI 25.2, 42.8, *p* < 0.01). There was an improvement in Achilles structure with mean organised echo pixels (UTC type I+II, in %) increasing from 79.9 (±11.5) to 86.4 (±10.0) (mean difference = 6.5%, 95% CI 0.80, 13.80, *p* =0.01), exceeding the 3.4% minimum detectable change. All 16 patients reported satisfaction with the procedure and 14 returned to pre-injury activity levels. There were no reported complications. Conclusions: Improved pain, function and tendon structure were observed 24 months after treatment with Achilles scraping and plantaris excision. The improvement in structure on the medial side of the Achilles after plantaris removal indicates that compression from the plantaris tendon might be an important presenting factor in this sub-group.

## 1. Introduction

Achilles tendinopathy is defined as a pathological process affecting the Achilles tendon, characterised by clinical symptoms that consist of load-related tendon pain, increased stiffness and functional impairment [1]. Recent prevalence studies report that Achilles tendinopathy is common in the general population [2,3,4] and the prevalence is higher among sport active individuals [5,6]. The first line approach for persistent and painful midportion Achilles tendinopathy is conservative management [7]. Loading regimens such as eccentric training and heavy slow resistance have demonstrated good clinical results in a majority of patients [7,8,9,10,11], but not all respond positively to rehabilitation [12]. While sub-optimal response to loading is hypothesised to be multi-factorial, a sub-group of non-responders are suspected to have midportion Achilles tendinopathy complicated by plantaris tendon involvement [13,14].

Results from recent studies infer that the plantaris tendon can under certain circumstances interfere with the Achilles tendon, favouring the development of tendinopathy through shearing or compressive forces [15,16]. The occurrence and degree of interaction are thought to be influenced by the anatomic positioning of the plantaris tendon, which is known to vary among individuals, sometimes even invaginating into the medial aspect of the Achilles [17,18]. Clinically, patients commonly report medial Achilles pain and exhibit tendinopathic tissue changes in the medial region of the Achilles tendon [19]. Histopathological and immunohistochemical studies have confirmed tendinopathic features in these closely appositioned plantaris tendons, and a peritendinous tissue with a comparably high degree of fat, sensory innervation and inflammatory changes between the plantaris and Achilles tendons [20,21,22]. Biomechanical studies also demonstrate high pressure between the plantaris and Achilles tendons with increased pressure occurring at end-range loading [16].

Developing successful exercise protocols for this patient sub-group has been problematic [13,14]. Previous case series have demonstrated short and longer-term improvement in pain and functional scores after surgical plantaris removal together with local Achilles scraping [13,23,24,25,26]. Furthermore, short-term studies monitoring post-operative Achilles tendon structure using Ultrasound Tissue Characterisation (UTC) have reported a return towards the expected alignment of the collagen structure in the medial Achilles 6–12 months after plantaris excision [25,26]. No studies to date have analysed longer-term outcomes and safety for this procedure. The aim of this study was to examine longer-term (two years) outcomes for pain, function and tendon structure after plantaris excision and local Achilles scraping in patients with chronic painful Achilles tendinopathy, medial side tenderness and suspected plantaris tendon interference. The hypothesis was that there is a significant improvement in VISA-A scores and improvement of tendon structure on UTC 24 months after surgical intervention.

## 2. Materials and Methods

### 2.1. Study Design and Inclusion Criteria

A prospective case series of consecutive patients with mid-portion Achilles tendinopathy and plantaris tendon involvement was included in this study. Patients were evaluated at Pure Sports Medicine in London, with subsequent procedures performed at the HCA Princess Grace Hospital in London. All patients were recruited between August 2014 and October 2015.

All patients had had a long duration of pain symptoms from the Achilles midportion (mean duration 28 months), and had tried rest and different types of exercise/loading treatments without success. Furthermore, three patients had tried cortisone injection therapy and five underwent shockwave therapy.

At clinical assessment, all patients reported pain at the medial aspect of the Achilles midportion during tendon loading activities, and tenderness at the ventromedial side of the Achilles midportion. Ultrasound plus Doppler (US+CD) and UTC examination verified localised tendinopathic changes at the ventral and medial aspects of the Achilles midportion, and a plantaris tendon located close to the medial aspect of the Achilles midportion. Patients who did not meet these clinical and imaging criteria were excluded and offered alternative treatment.

Given the requirement for long-term follow-up, patients needed to be living or working in the UK at the time of initial assessment to be included in the study.

### 2.2. Surgical Treatment

Patients were surgically treated with US-guided local Achilles scraping plus plantaris tendon excision, performed under local anaesthesia by the same surgeon (HA). The medial aspect of the tendon was visualised via a short longitudinal incision on the medial side of the Achilles tendon midportion. A thickened plantaris tendon in close relationship to the medial aspect of the Achilles tendon was identified in all patients (see Figure 1). Surgical treatment consisted of a release of the plantaris tendon followed by excision of the plantaris tendon distally close to the calcaneal insertion and proximally at a level slightly above the distal medial soleus muscle insertion (see Figure 1). In addition, a local Achilles tendon scraping procedure releasing the richly vascularised fat tissue from the Achilles tendon was performed using a scalpel in the regions with tendinopathy, and ultrasound+Doppler verified high blood flow on the ventromedial side of the Achilles tendon.

### 2.3. Post-Operative Rehabilitation

Post-operative rehabilitation consisted of a range of movement exercises, full weight bearing loading from day two after surgery, and a structured strengthening program (for more details, see [25]). Adverse events were recorded by the main investigator (LM) and included infection, wound breakdown and requirements for further treatment(s).

### 2.4. Follow-Up Evaluation

Evaluation was performed at 24 months post-procedure. Patient satisfaction with the result of the operation was evaluated by dichotomous self-reporting (satisfied or not satisfied). VISA-A scores [27] were used to assess pain and function and UTC was used to assess tendon structure. UTC scanning was performed in a standardised position as previously described [19,26]. UTC algorithms (UTC 2010, UTC Imaging) were used to quantify the dynamics of grey levels of corresponding pixels in contiguous images over 17 images [28,29]. UTC algorithms can discriminate 4 different echo types (types I to IV). In this study, echo types I and II were grouped together as they represented organised structure, whereas echo types III and IV represent disorganised structure. The echo structure of the tendon was quantified from the insertion to the soleus musculotendinous junction and expressed as a percentage of tendon volume. This method has previously been reported to be reliable and have a minimal detectable difference of 3.4% [30].

### 2.5. Statistics

Analysis was performed using SPSS (v.20 for Windows, SPSS Inc., Chicago, IL, USA). A two-tailed, paired samples *t*-test (parametric) was used to compare pre-operative and post-operative VISA-A scores with associated 95% confidence intervals and mean change reported. A Wilcoxon signed rank test (non-parametric) was used to assess the UTC scan results. A *p*-value of 0.05 defined statistical significance.

### 2.6. Ethics

Ethical approval was granted by Umea University, Umea, Sweden (DNR 98-35). Patients gave written informed consent at the time of recruitment.

## 3. Results

Eighteen patients (13 males and 5 females) with a mean age of 39.2 years (±7.2 years) and a mean symptom duration of 27.9 months (±34.9 months) were eligible for inclusion (see Table 1). All patients were active in sports, including four in track and field, one in football, one in cricket, one in rugby and eleven in endurance running. Three patients were elite athletes.

Sixteen patients (89%) were evaluated at 24 months, with two patients who could not be contacted lost to follow-up. All 16 patients reported satisfaction with the result of the operation, and 14 had returned to pre-injury sport activity levels. There were no reported complications.

Mean VISA-A scores increased from 58.2 (±15.9) to (92.0 ± 9.2) (mean difference 33.7, 95% CI 25.2, 42.8, *p* < 0.01), exceeding the minimum clinically important difference of 10 points used in previous studies and systematic reviews [31,32]. Individual VISA-A responses can be viewed in Figure 2. The structure of the Achilles mid-portion improved, with organised echo pixels (UTC type I + II) increasing from 79.9 (±11.5) to 86.4 (±10.0) (mean difference 6.5%, 95% CI = −0.80–13.80, *p* = 0.01), exceeding the 3.4% minimum detectable change [30] (see Table 2 and Figure 3).

## 4. Discussion

This prospective case series aimed to examine longer-term (24 month) outcomes for pain, function and tendon structure after plantaris excision and local Achilles scraping in patients with chronic painful Achilles tendinopathy, reporting good clinical outcomes and significant structural improvement.

Positive short-term clinical outcomes after localised Achilles tendon scraping and plantaris tendon excision have been demonstrated in previous studies [22,23,24,25], but to the best of our knowledge this is the first study to demonstrate good clinical outcomes together with the improvement of Achilles tendon structure at 24 months after surgical treatment. In a previous study, decreased tendon thickness and improved structure verified by grey-scale ultrasound were reported after successful sclerosing polidocanol injection treatment in patients with chronic painful midportion Achilles tendinopathy [33]. In the current study UTC was used, which is a more reliable and objective method to analyse changes in tendon structure [30].

Recent research has demonstrated that plantaris tendon involvement can be found in patients with chronic midportion and insertional Achilles tendon tendinopathy, especially when there is medial Achilles tendon pain [19,34,35,36]. Cadaveric studies have demonstrated that there is an individual variability of the course of the plantaris tendon in relation to the Achilles tendon, with up to nine different courses demonstrated [17]. The plantaris tendon is reported to be stronger and stiffer than the Achilles tendon [37], and recent cadaveric studies have demonstrated that the plantaris tendon has a tri-planar movement pattern in relation to the Achilles tendon [15]. It is hypothesised that the variability in the course of the plantaris tendon, coupled with the differences in mechanical properties of the tendons, may lead to compressive and/or shearing forces on the medial Achilles. Studies on intrinsic and extrinsic foot muscle morphology have found significant differences in cross-sectional areas between healthy individuals and patients with midportion Achilles tendinopathy [38,39]. These results further support that chronic pain conditions can be linked to changes in biomechanical foot and ankle biomechanics potentially causing Achilles tendon disturbance. The compression hypothesis between the Achilles and plantaris tendon is further supported by a recent cadaveric study demonstrating increased compressive forces between the two tendons at end-range plantarflexion [16]. The excision of the plantaris tendon could therefore lead to reduced compressive forces on the medial Achilles with resultant improvement in symptoms and tendon structure [26]. The results of the present study demonstrate that the short-term improvements reported in previous studies [24,26] can be maintained even 24 months after the surgical intervention.

It is likely that different pathways can play a role in a certain sub-group of patients. In a recent study, it was demonstrated that in some patients with medial Achilles pain there is plantaris tendinopathy alone together with a structurally normal Achilles tendon [40]. Furthermore, histological studies have demonstrated comparably high degrees of sensory innervation within the plantaris tendon, highlighting its potential role as a pain mediator [21]. In contrast to these findings, research from Calder and colleagues reports that in some patients surgical plantaris removal leads to good clinical results even in the absence of pronounced plantaris tendinopathy [22]. This highlights the likely role of the biomechanical interference between the two tendons and the pain driving potential of the interpositioned peritendinous tissue, which is characterised by high degrees of sensory innervation and inflammatory changes [21,22]. In previous studies using the Achilles scraping procedure alone, cases that failed surgery and required a re-operation had a thickened plantaris located in close apposition to the medial Achilles tendon [13,14]. These failed cases eventually improved with the removal of the plantaris tendon. As exercise approaches so far have not been successful for the treatment of patients with diagnosed plantaris tendon involvement, it seems that surgical intervention is needed to allow for a return to pain-free Achilles tendon loading in this sub-group. As such, recent studies have begun to explore better diagnostic strategies and use less invasive ultrasound-guided percutaneous surgical treatments [19,35,41,42].

A limitation of this study is the low sample size and the absence of a control group (or non-surgical cohort). It is therefore not possible to determine whether the Achilles scraping procedure, the plantaris removal, a combination, or other factors (such as natural history) are responsible for the improved pain and function and improved tendon structure identified. The prolonged duration of symptoms in most of the recruited patients and the previous unsuccessful application of conservative treatments mean that it is unlikely that natural history explains the observed outcomes. Improvements could theoretically be due to the rehabilitation protocol used after surgery, though immediate full weight bearing loading and a quick return to sports were used. Future higher-level randomised studies should be performed, including a comparative non-operative group, or a sham surgical group, to investigate the effectiveness of the surgical procedure in this sub-group of patients with mid-portion Achilles tendinopathy and plantaris involvement.

The clinical implications of this study add to our knowledge of the management of a sub-group of Achilles tendinopathy. Patients with mid-portion Achilles tendinopathy with clinical features of plantaris tendon interference should be considered for plantaris excision if conservative treatment fails. Short and long-term data now suggest good outcomes after plantaris excision. 

## 5. Conclusions

This prospective case series demonstrates good clinical outcomes and significant structural improvement in the Achilles tendon after scraping and plantaris tendon removal in a sub-group of patients with chronic midportion Achilles tendinopathy and plantaris involvement.

## Figures and Tables

**Figure 1 jcm-10-02695-f001:**
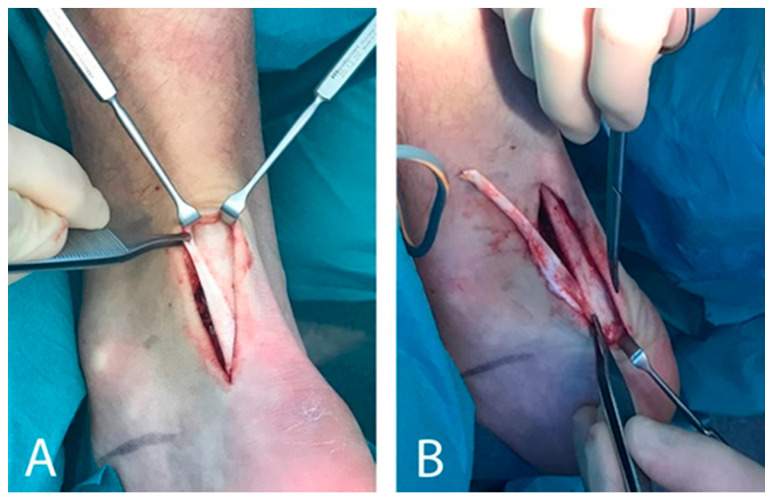
A wide and thick plantaris tendon (**A**) that is surgically released (**B**) from the medial side of the Achilles midportion.

**Figure 2 jcm-10-02695-f002:**
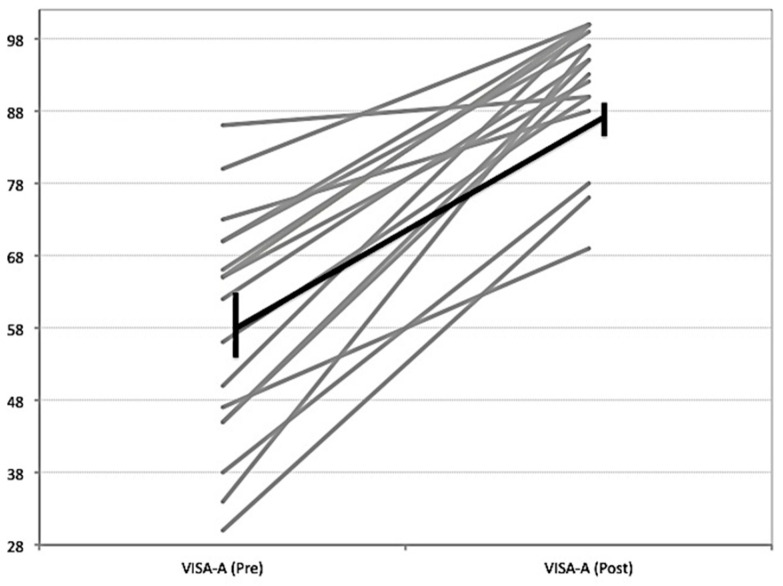
Individual VISA-A responses pre- and post-surgery (24 months) (in grey). The average improvement is indicated by the black line.

**Figure 3 jcm-10-02695-f003:**
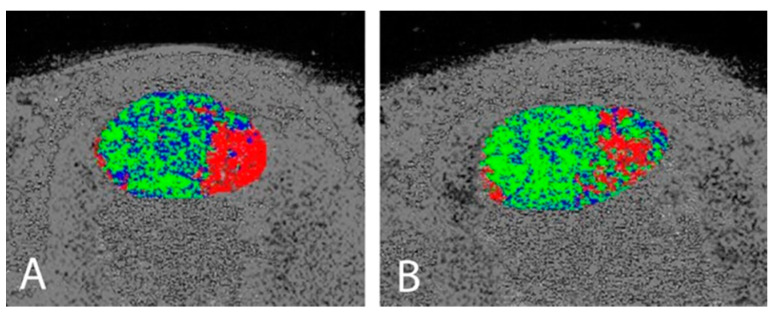
Pre- (**A**) and post-operative (**B**) UTC examinations demonstrating an increase in organised (green) tissue in the post-operative (**B**) Achilles tendon.

**Table 1 jcm-10-02695-t001:** Demographics of patients.

Patients	
**Male/female**	13/5
Age (Mean/SD) (Months)	39.2 (±7.2)
Symptom duration (Mean/range) (Months)	27.9 (2–108)
Baseline VISA-A (Mean/SD)	58.2 (±15.9)
Elite/amateur	3/15

Key: VISA-A = Victorian institute of sports assessment-Achilles; SD = standard deviation.

**Table 2 jcm-10-02695-t002:** Results of VISA-A and UTC.

	Visa A Mean (±)	Echo Type I + II Mean (±)
Pre-surgery	58.2 (15.9)	79.9 (11.5)
24 months post-surgery	92.0 (9.2)	86.4 (10.0)
Mean difference	33.8 (95% CI 25.2, 42.8) *p* < 0.01	6.5% (95% CI 0.80, 13.80) *p* = 0.01

Key: VISA-A = Victorian institute of sports assessment-Achilles; UTC = ultrasound tissue characterisation.

## Data Availability

The data presented in this study are available on request from the corresponding author.

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
