# Peer review of "Achilles Scraping and Plantaris Tendon Removal Improves Pain and Tendon Structure in Patients with Mid-Portion Achilles Tendinopathy—A 24 Month Follow-Up Case Series"

_jcm, 2021, doi:10.3390/jcm10122695_

Round 1
Reviewer 1 Report
This study analyzed the efficacy of Achilles Scraping and Plantaris Tendon Removal in Patients with Resistant Mid-portion Achilles Tendinopathy and Medial Achilles Pain. The follow up period was 2 years. Regarding the follow up period, it is relatively interesting paper.
However, 2 years is not that long period. Also this procedure and this type of report is not novel.
Although the author described the well structured-operative treatment, they need the control using non-operative treatment. This is critical point.
Also this is the prospective study, so they need to control the experimental group. They included the elite and non-elite athletics. There was no information about the past treatment history including injection, ESWT and so on.
If they can provide the result of another time point except 24 mo like 6 mo, 12 mo, it will be great.
Author Response
We thank the reviewer for his valuable comments.
This study analyzed the efficacy of Achilles Scraping and Plantaris Tendon Removal in Patients with Resistant Mid-portion Achilles Tendinopathy and Medial Achilles Pain. The follow up period was 2 years. Regarding the follow up period, it is relatively interesting paper.
Thank you.
However, 2 years is not that long period. Also this procedure and this type of report is not novel.
Thank you for this comment. According to Cochrane guidelines a long term period is defined as >1 year. However, we agree that longer follow up periods e.g. 5 years would be even better and we will try to do this later on. We have consistantly used the term „longer-term“ follow up now. There had been studies by us on the same procedure, but this ist he first study on longer-term results (pain AND tendon structure) on surgically treated patients with plantaris tendon involevement.
Although the author described the well structured-operative treatment, they need the control using non-operative treatment. This is critical point.
Thank you. We agree. A control group using non-operative treatment would be ideal, but for patients that already have had a long duration of symptoms not responding to non-operative treatment this is not an alternative, and it could be questioned from an ethical point of view. However, we have discussed this as limitation in the discussion.
Also this is the prospective study, so they need to control the experimental group. They included the elite and non-elite athletics.
We agree that this is a limitation and therefore we have discussed the absence of a control and the design as case series clearly in the discussion.
There was no information about the past treatment history including injection, ESWT and so on.
Thank you. We only had written on the loading history. Now we have also added information on other teratment types.
„All patients had had a long duration of pain symptoms from the Achilles midportion, had tried rest and different types of exercise/loading treatments without success. Furthermore, three patients had tried cortisone injection therapy and five underwent shockwave therapy.“
If they can provide the result of another time point except 24 mo like 6 mo, 12 mo, it will be great.
That would be indeed be great, Unfortunately we don’t have follow up data for all these patients at 6 and 12 months as there were several elite athletes that were not available. However, we have published the results for some in a previous study (Ref 26). The conclusion in this current study ist hat this short-term improvement (Ref26) can be maintained over a longer period.
Reviewer 2 Report
Thank you for opportunity for reviewing this interesting paper. The research adhere to reporting observational guidelines. After carefully reading this manuscript, I must say that, from my point of view, the authors have done research on an important topic related with the achilles scraping and plantaris tendon removal improves pain and tendon structure in patients with resistant mid-portion achilles tendinopathy and medial achilles pain. This could be interesting clinicians, universities, private research organizations, and independent scientists, that frequently work in this area. It could give them a wider concept about and helps advance recognition of the input of different health professionals into the management of this condition, and helps inform the need for further multi-professional work in this area.
This is an interesting aim with the are characteristics of the achilles scraping and plantaris tendon removal improves pain and tendon structure in patients with resistant mid-portion achilles tendinopathy and medial achilles pain. I have considered the quality of the manuscript redaction and presentation, the quality of the research methodology, the novelty and importance of the observations, and the appropriateness for the Journal’s readers (according with the Journal’s name) and I think that this manuscript joins adequate conditions to be accepted for publication JCM.
I have no real problems with the text of this paper, only some suggestions that are mentioned below. It appears as if the authors have done the study well and have answered an interesting clinical question with their work.
Also, there are a major concerns with the manuscript that require attention prior to publication. These will be discussed below relative to the sections of the manuscript.
TITLE
The title of this manuscript is very long. Perhaps a more concise version for clarity, interes and ease of read.
KEYWORDS:
Please use recognised MeSH terms as this will assist others when they are searching for information on your research topic. The following website will provide these (simply start typing in a keyword and see if it exists or find an alternative if it does not): https://www.ncbi.nlm.nih.gov/mesh
INTRODUCTION
I suggest that background should be improved, with more details about the importance of plantaris tendon and mid-portion achilles tendinopathy and medial achilles pain.
It is indeed important paper but it lacks several critical references, in which it was presented related with this condition, and it should be emphasized in the INTRODUCTION or Discussion of the authors' paper. More info info in
Intrinsic foot muscles morphological modifications in patients with Achilles tendinopathy: A novel case-control research study
https://pubmed.ncbi.nlm.nih.gov/31593918/
Ultrasound evaluation of extrinsic foot muscles in patients with chronic non-insertional Achilles tendinopathy: A case-control study
https://pubmed.ncbi.nlm.nih.gov/30844628/
METHODS
This section is poor, needs to present a better rationale for the study and the methodology employed. Also, neither appear information related with inclusion and exclusion criteria, dates, protocol.
Likewise more detail about information calculate sample size and data should be provided. Also, please need include the data and record code and all information related with the ethics committee and explain aspects ethics and legal requirement about this research.
RESULTS
The results is clear and concise with appropriate statistical analysis been performed appropriately and rigorously.
DISCUSSION
Please to chage the word Conclusion by Discussion. Also, to include this section the principal strengths and weaknesses in relation to other studies, discussing important differences in results; the meaning of the study: possible explanations and implications and unanswered questions and future research
CONCLUSION:
Summarize the conclusions in order to reflect only the study findings.
Author Response
We thank the reviewer for his valuable comments.
Thank you for opportunity for reviewing this interesting paper. The research adhere to reporting observational guidelines. After carefully reading this manuscript, I must say that, from my point of view, the authors have done research on an important topic related with the achilles scraping and plantaris tendon removal improves pain and tendon structure in patients with resistant mid-portion achilles tendinopathy and medial achilles pain. This could be interesting clinicians, universities, private research organizations, and independent scientists, that frequently work in this area. It could give them a wider concept about and helps advance recognition of the input of different health professionals into the management of this condition, and helps inform the need for further multi-professional work in this area.
Thank you.
This is an interesting aim with the are characteristics of the achilles scraping and plantaris tendon removal improves pain and tendon structure in patients with resistant mid-portion achilles tendinopathy and medial achilles pain. I have considered the quality of the manuscript redaction and presentation, the quality of the research methodology, the novelty and importance of the observations, and the appropriateness for the Journal’s readers (according with the Journal’s name) and I think that this manuscript joins adequate conditions to be accepted for publication JCM.
I have no real problems with the text of this paper, only some suggestions that are mentioned below. It appears as if the authors have done the study well and have answered an interesting clinical question with their work.
Thank you.
Also, there are a major concerns with the manuscript that require attention prior to publication. These will be discussed below relative to the sections of the manuscript.
TITLE
The title of this manuscript is very long. Perhaps a more concise version for clarity, interes and ease of read.
We have now changed the title and made it shorter.
„Achilles scraping and plantaris tendon removal improves pain and tendon structure in patients with mid-portion Achilles tendinopathy – a two-year follow-up case series“
KEYWORDS:
Please use recognised MeSH terms as this will assist others when they are searching for information on your research topic. The following website will provide these (simply start typing in a keyword and see if it exists or find an alternative if it does not): https://www.ncbi.nlm.nih.gov/mesh
Thank you for this helpful advice. We have now changed into the following keywords:
„Achilles tendon“, „tendinopathy“; „plantaris“; „surgery“; „after treatment“; „follow up study“
INTRODUCTION
I suggest that background should be improved, with more details about the importance of plantaris tendon and mid-portion achilles tendinopathy and medial achilles pain.
We don’t agree. We have ourself done plenty of research studies on the plantaris tendon and its importance in a subgroup of patients with medial Achilles tendon pain. We know for sure that we have put all the information that is curently known in either the introduction or the discussion part.
It is indeed important paper but it lacks several critical references, in which it was presented related with this condition, and it should be emphasized in the INTRODUCTION or Discussion of the authors' paper. More info info in
Intrinsic foot muscles morphological modifications in patients with Achilles tendinopathy: A novel case-control research study
https://pubmed.ncbi.nlm.nih.gov/31593918/
Ultrasound evaluation of extrinsic foot muscles in patients with chronic non-insertional Achilles tendinopathy: A case-control study
https://pubmed.ncbi.nlm.nih.gov/30844628/
Thank you. We have read these articles and have integrated them into the discussion.
METHODS
This section is poor, needs to present a better rationale for the study and the methodology employed. Also, neither appear information related with inclusion and exclusion criteria, dates, protocol.
We agree. We have now added sub-headings and added the missing information.
Likewise more detail about information calculate sample size and data should be provided.
As this is a case series with no control group a sample size calculation is not needed. The weakness of having a case series has been discussed. As all patients had tried other treatments before (e.g. all tried loading regimens) inclduing a control group on conservative treatment could be questioned from an ethical point of view.
Also, please need include the data and record code and all information related with the ethics committee and explain aspects ethics and legal requirement about this research.
Thank you. We already had this information in the text. However, to make it more clear we have included subheadings now.
RESULTS
The results is clear and concise with appropriate statistical analysis been performed appropriately and rigorously.
Thank you.
DISCUSSION
Please to chage the word Conclusion by Discussion.
Done.
Also, to include this section the principal strengths and weaknesses in relation to other studies, discussing important differences in results; the meaning of the study: possible explanations and implications and unanswered questions and future research
Done.
CONCLUSION:
Summarize the conclusions in order to reflect only the study findings.
We agree. Done.
Reviewer 3 Report
Glad to have an opportunity to review this manuscript, There are severe problematic areas of the
manuscript and the authors were not able to deal with the essential aspects of so-called “scientific
research.”
There are several methodological concerns that limit the readers understanding of why this
experiment was conducted. Below I have provided comments for the authors.
This study cannot be progressed into any further steps of publication in this quality journal unless
the following issues properly dealt with:
Even though the abstract is successfully compiling and summarizing focal points of this study
Validity issue – your study is successful to provide validity evidence of your measurement issues. I
am convinced to your findings are clear about what would be potential lessons from reading your
study.But, could you explain the benefits of your achievement on clinical situacions?
Introduction section may be improved adding new information in order to provide an adequate state-
of-the-art including some references.
There are not good description of the properties of the outcome measurements as a detailed
statistical analyses In fact, Tables, figures and redaction of the results do not shows the main finding of the
study. Even though authors have included new tables after the first send I suggest to author to include significance levels in tables in order to improve their results
Please provide a hypothesis.
Discussion section may include future research studies secondary to the current findings of this
study. Clinical considerations, limitations and overall discussion are well-presented, but future
research may be useful in order to propose future research regarding this field.
However I suggest to authors should include some reference related to prior studies, in this section
for example in back disorders
I suggest include the following references to complete the requeriment
-doi: 10.3390/jcm10081793
Or in the case of climientric tools:
-doi: 10.3390/ijerph15102205
Please check and re-confirm and have other experienced scholars to read your manuscript prior to“submission” in terms of “research process” and conclusion part.
Author Response
We thank the reviewer for his valuable comments.
Glad to have an opportunity to review this manuscript, There are severe problematic areas of the
manuscript and the authors were not able to deal with the essential aspects of so-called “scientific
research.”
There are several methodological concerns that limit the readers understanding of why this
experiment was conducted. Below I have provided comments for the authors.
This study cannot be progressed into any further steps of publication in this quality journal unless
the following issues properly dealt with:
Even though the abstract is successfully compiling and summarizing focal points of this study
Validity issue – your study is successful to provide validity evidence of your measurement issues.
I am convinced to your findings are clear about what would be potential lessons from reading your
study.But, could you explain the benefits of your achievement on clinical situacions?
We have now added a paragraph in the end of the discussion to highlight the clinical benefits of this study.
Introduction section may be improved adding new information in order to provide an adequate state-
of-the-art including some references.
We don’t agree. We have done plenty of research on the plantaris tendon and its importance in a subgroup of patients with medial Achilles tendon pain. We know for sure that we have put all the information that is curently known in either the introduction or the discussion part.
There are not good description of the properties of the outcome measurements as a detailed
statistical analyses
We agree and have changed it.
In fact, Tables, figures and redaction of the results do not shows the main finding of the study.
We don’t agree. The main results are stated in Table 2, Figure 2+3 in which the improvement of VISA-A scores and tendon structure (UTC) can be seen.
Furthermore, Figure 1 is an illustration of the surgical process which is helpful for understanding the process and in Table 1 there are demographic data from the patients. We believe that all these informations are essential to report.
Even though authors have included new tables after the first send I suggest to author to include significance levels in tables in order to improve their results.
Done.
Please provide a hypothesis.
Done.
„The hypothesis was that there is significant reduction of VISA-A scores and improvement of tendon structure on UTC 24 months after surgical intervention.“
Discussion section may include future research studies secondary to the current findings of this
study. Clinical considerations, limitations and overall discussion are well-presented, but future
research may be useful in order to propose future research regarding this field.
We have already included future research aspects:
„Future higher-level randomized studies should be performed, including a comparative non-operative group, or a sham surgical group, to investigate the effectiveness of the surgical procedure in this sub-group of patients with mid-portion Achilles tendinopathy and plantaris involvement.“
However I suggest to authors should include some reference related to prior studies, in this section
for example in back disorders
I suggest include the following references to complete the requeriment
-doi: 10.3390/jcm10081793
Or in the case of climientric tools:
-doi: 10.3390/ijerph15102205
Please check and re-confirm and have other experienced scholars to read your manuscript prior to
“submission” in terms of “research process” and conclusion part.
We have discussed this with other peers. We have made changes in the discussion/conclusion part. Concerning the design we have discussed this weakness in the limitations (case series).
Round 2
Reviewer 1 Report
Abstract:
The aim of this case series was to determine the long-term outcomes on pain and Achilles tendon structure.
-> This sentence should be modified. It is not clear.
compression from the plantaris tendon might be an important presenting factor in this sub-group.
-> (Compression) Actually, this word is not determined in this study.
Key words
Consider delete "after treatment"
Introduction
The hypothesis was that there is significant reduction of VISA-A scores and improvement of tendon 79
structure on UTC 24 months after surgical intervention.
-> Reduction? Is this correct?
Participants
The duration of symptom was 2 - 108 mo.
Results
Eighteen patients (13 males and 5 females) with a mean age of 39.2 years (±7.2 years) and a mean symptom duration of 27.9 months (±34.9 months) (range 2-108) met the inclusion criteria and were included were eligible for inclusion (see table one1). All patients were active in sports including, four in track and field, one in football, one in cricket, one in rugby and eleven in endurance running. Three patients were elite athletes.
-> These sentences should be in the M and M.
It will be great if they can provide the subgroup analysis (Cortisone injection + vs - or ESWT + vs -)
Also it will be great if they can provide the histologic change of excised tendon comparing the plantaris and achilles.
Final number of participants is only 16. So it will be great if they can provide the patient data in detail. One by one.
Reviewer 2 Report
In their first revision of manuscript, the authors have addressed my questions/comments properly.
Reviewer 3 Report
I want to thank you for the opportunity to review this manuscript. After the review and in my humble opinion, from my point of view, the manuscript presents major problems that inaceptable for publication:
In their first revision of manuscript, the authors have not addressed my questions/comments properly.
Even though I had indicated exactly what should they do, authors have not completed my requirement for this reason I suggest they need to improve their manuscript in order to increase the manuscript quality
For example, I do not why they have not included the references with regard
Back disorders:
-doi: 10.3390/jcm10081793
Or in the case of climientric tools:
-doi: 10.3390/ijerph15102205